# Innovative Solutions for Drought: Evaluating Hydrogel Application on Onion Cultivation (*Allium cepa*) in Morocco

**Omnia El Bergui** [1,2]**, Aziz Abouabdillah** [1,*]**, Mohamed Bourioug** [1]**, Dominik Schmitz** [3]**, Markus Biel** [3]**, Abdellah Aboudrare** [1]**, Manuel Krauss** [4]**, Ahlem Jomaa** [4]**, Sebastian Romuli** [5]**, Joachim Mueller** [5]**, Mustapha Fagroud** [1] **and Rachid Bouabid** [1]

1. Department of Agronomy and Plant Breeding, Ecole Nationale d'Agriculture de Meknès, Meknes 50001, Morocco; oelbergui@enameknes.ac.ma (O.E.B.); mbourioug@enameknes.ac.ma (M.B.); aaboudrare@enameknes.ac.ma (A.A.); mfagroud@enameknes.ac.ma (M.F.); rbouabid@enameknes.ac.ma (R.B.)
2. Departement of Agronomic and Agri-Food Science, Institut Agronomique et Vétérinaire Hassan II, Rabat 10101, Morocco
3. IAP—Institute for Applied Polymer Chemistry, University of Applied Sciences Aachen, 52428 Aachen, Germany; dominik.schmitz@fh-aachen.de (D.S.); biel@fh-aachen.de (M.B.)
4. Research Institute of Water Management and Climate Future, RWTH Aachen University, 52062 Aachen, Germany; krauss@fiw.rwth-aachen.de (M.K.); jomaa@fiw.rwth-aachen.de (A.J.)
5. Institute of Agricultural Engineering, University of Hohenheim, 70599 Stuttgart, Germany; sebastian_romuli@uni-hohenheim.de (S.R.); joachim.mueller@uni-hohenheim.de (J.M.)
* Correspondence: aabouabdillah@enameknes.ac.ma; Tel.: +212-620-470-152

**Abstract:** Throughout the last decade, and particularly in 2022, water scarcity has become a critical concern in Morocco and other Mediterranean countries. The lack of rainfall during spring was worsened by a succession of heat waves during the summer. To address this drought, innovative solutions, including the use of new technologies such as hydrogels, will be essential to transform agriculture. This paper presents the findings of a study that evaluated the impact of hydrogel application on onion (*Allium cepa*) cultivation in Meknes, Morocco. The treatments investigated in this study comprised two different types of hydrogel-based soil additives (Arbovit® polyacrylate and Huminsorb® polyacrylate), applied at two rates (30 and 20 kg/ha), and irrigated at two levels of water supply (100% and 50% of daily crop evapotranspiration; *ETc*). Two control treatments were included, without hydrogel application and with both water amounts. The experiment was conducted in an open field using a completely randomized design. The results indicated a significant impact of both hydrogel-type dose and water dose on onion plant growth, as evidenced by various vegetation parameters. Among the hydrogels tested, Huminsorb® Polyacrylate produced the most favorable outcomes, with treatment T9 (100%, HP, 30 kg/ha) yielding 70.55 t/ha; this represented an increase of 11 t/ha as compared to the 100% *ETc* treatment without hydrogel application. Moreover, the combination of hydrogel application with 50% *ETc* water stress showed promising results, with treatment T4 (HP, 30 kg, 50%) producing almost the same yield as the 100% *ETc* treatment without hydrogel while saving 208 mm of water.

**Keywords:** hydrogel; deficit irrigation; onion; yield; water economy

## 1. Introduction

Water resources in Morocco are becoming increasingly rare due to the spatio-temporal irregularity of rainfall, the rising demographic pressure, and the development of the agricultural, tourism, and industrial sectors. In this climatic and socio-economic context, Morocco must engage in a strategic rational use of its water resources and optimize their consumption [1].

As the future projections of the Mediterranean climate in general [2] and of Morocco's climate in particular highlight the great threat of drought [3], it has become a real threat to

the country's food security and sovereignty, as well as the availability of water resources in rural and urban areas.

Agriculture is the major water consumer and is estimated to use about 80% of the world's resources, especially for irrigation [4], as it is a limiting factor for the survival, growth, and productivity of crops [5]. This sector represents nearly 15% of Morocco's GDP (gross domestic product) and engages nearly 40% of the labor force.

The Fez-Meknes region is one of the 12 regions of the country according to the 2015 administrative division. With an area of 40,075 km$^2$ and a total population of 4.23 million inhabitants, it contributes to the national GDP with 9.39%. Economically, the region holds 15% of the national useful agricultural area, 15% of the national cereal production, and 14% of the national forests [6].

The climate of the region ranges from Mediterranean to continental with hot winters and summers, especially in the province of Boulemane. The high areas of the Rif and Pre-Rif have a warm climate in summer, whereas in winter it is colder with frequent and severe frosts. The continental zones are subject to the blows of the 'Chergui' which contribute to the rise in temperature [6]. As a result of its geographical disparities, the region of Fez-Meknes has three main categories of zones: the wet zone that receives a significant amount of annual rainfall (ranging from 600 mm/year to more than 800 mm/year); followed by the medium rainfall zone, i.e. the Pre-Rif and the northern flank of the region (our study site), which receives an average of 400 mm of rainfall per year; and finally the dry zone, which are the alfa areas in the southeast of the region that receive annual precipitation of less than 300 mm.

With the world and Morocco facing these major water shortages and drought challenges, farmers have started to look for innovations and cultural practices to improve water efficiency as well as crop yields.

Hydrogels have emerged as an effective technique to enhance the soil's water-holding capacity and conserve moisture, particularly in arid and semi-arid areas [7]. These hydrophilic gels are crosslinked materials that absorb water without dissolving. They increase water availability in the soil for plants, minimize wastage due to evaporation and percolation, and improve crop growth and yield parameters [8].

Recent studies have highlighted the benefits of hydrogels as hydrophilic polymers that improve soil porosity, water holding capacity [9], aeration, infiltration, nutrient uptake [10], and water absorption to promote plant growth (e.g., [11]).

In Morocco, hydrogels have been noted to have a highly significant effect on plant growth, contributing to remarkable height growth of argan plants from the fourth month after planting [12]. Additionally, Tyagi et al. [13] and Hou et al. [14] observed a stimulating hydrophilic superabsorbent effect on the yield of different crops, such as potatoes and maize, under various soil conditions

According to Johnson [15], the use of hydrogels increases available moisture in the root zone, which implies longer intervals between irrigations. Furthermore, in a study conducted by Satriani et al. [16], it was found that irrigating at 70% of crop evapotranspiration demand and amending the soil with hydrogel maximized crop water productivity values and did not adversely affect yield for the bean crop.

In a similar vein, Barakat et al. [17], found that injecting 150 g per banana plant (Large Dwarf) while providing only 80% *ETc* to the plants saved 20% of water without affecting the growth and yield of the banana plant. Han et al. [18] found that 900 m$^3$/hm$^2$ of irrigation water could be saved when the hydrogel is applied at 200 g per apple tree. Moreover, in a greenhouse in Canada, Suresh et al. [19] tested the effect of Superab A200 (acrylamide-based synthetic hydrogel) on tomato yields when planted in 7.6 L pots filled with Agromix G6 (substrate containing a mixture of peat, coconut fiber, perlite, limestone, and gypsum). The yield increased with the presence of the hydrogel in the substrate, improving by 50% with a dose of 0.5%.

In spite of its many benefits, a major concern for water-retaining particles is their ability to biodegrade and prevent their absorption by the root and their eventual penetration into the fruit and plant material.

Following Decriaud-Calmon et al. [20], certain polymers can be degraded by five types of interacting mechanisms: photodegradation, chemical degradation (hydrolysis, oxidation), thermal degradation, mechanical degradation, and biodegradation in the presence of microorganisms depending on polymer type and structure.

The application of hydrogels represents an innovative solution to minimize micronutrient leaching into the groundwater and increase water use efficiency; in addition, it reduces the amount of fertilization, since nutrient leaching is avoided by reducing runoff.

On the other hand, hydrogels that contain fertilizers have a controlled release of water so that the dose of fertilizer is adjustable over time. The nutrient is available to the plant over a longer period of time rather than being available quickly, including ammonium nitrate, ammonium phosphate, or potassium chloride [21].

Accordingly, the aim of this research is to assess the response of onion (*Allium cepa*) crop to the combined effect of hydrogel application and deficit irrigation, while monitoring the growth, development, and eco-physiological and yield parameters under the climatic conditions of the Meknes region. Onion was selected since it is one of the most crucial and widespread crops globally. It is also a source of income for many farmers in Morocco, with an average annual planted area of 30,000 hectares. As a specific objective of this research, we aimed to find out the hydrogel's efficiency in improving or maintaining yields while reducing water by half, thus relying on the hydrogel soil's water retention and release characteristics to reduce drought stress.

## 2. Materials and Methods

### 2.1. Experimental Site

The study was carried out in a field located at the National School of Agriculture of Meknes (33°50′34″ N 5°28′24″ W, altitude 906 m), approximately 10 km southeast of the city of Meknes (Figure 1). The experimental plot had been in fallow mode for over 10 years prior to the study.

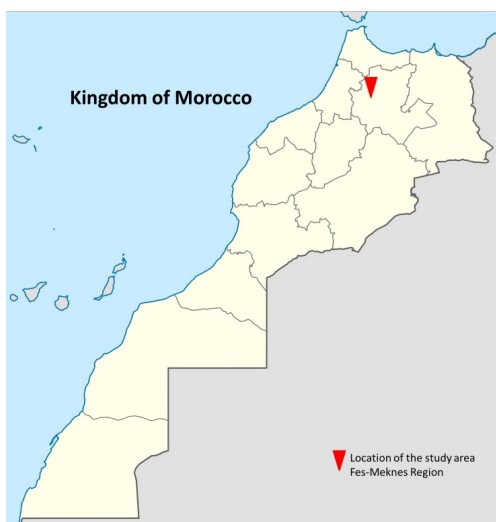

**Figure 1.** Location of the experiment site on Morocco's map, "Reprinted/adapted with permission from Ref [22]. 2023, OpenStreetMap contributors and Offnfopt".

Meknes city is situated at an altitude of 516 m above sea level and has a warm temperate climate characterized by winter rainfall. The average annual precipitation in the region is 509 mm, while the average annual temperature is 17.4 °C.

The soil in the study area is a clayey-silt soil, rich in organic matter with a neutral pH. The physical soil analysis revealed that the volumetric moisture at field capacity (HCC) was 38.68%, and the volumetric moisture at wilting point (HPFP) was 23.29%.

*2.2. Design and Layout*

The experimental material investigated comprised locally sourced red variety onion crop (*Allium cepa*). On 6 May 2021, the crop was transplanted at a density of $15 \times 20$ cm, with a depth from 1 to 2 cm, after being dipped in a $10$ mL/m$^2$ dose of a fungicide containing 722 g/L Propamocarb-HCl.

To ensure proper application of the hydrogels in each treatment, they were manually spread on the surface of the experimental plot, and then buried at a depth of 18–20 cm by plowing. The soil was then harrowed to level the surface.

Two types of hydrogels were used in the experiment:

- Huminsorb®:

It is an organic–mineral potassium fertilizer (16.4% $K_2O$ potassium oxide) made of plant materials (humic acid), clay minerals (bentonite), and synthetic polymers (crosslinked). It is a co-polymer of acrylic acid and potassium acrylate ($C_3H_3K_2O$, partially neutralized with crosslinked acrylic acid);

- Arbovit®:

It is an organic–mineral fertilizer (NK 4.2–14.11), which contains clay minerals (bentonites) and synthetic polymers. The hydrogel in the mixture is the same as in Huminsorb®.

Both types of hydrogels are storable products under dry and sunny conditions at a temperature above 6 °C and below 30 °C, being applied for planting at 2–3 g/m$^2$ (20–30 kg/ha as applied in this setup). On the surface, the products should be sprinkled evenly and buried deep in the root zone.

The hydrogel-based soil additives were characterized regarding swelling properties (maximum swelling degree $S_{max}$, kinetic, absorbency under load (AUL)), residual from the eluate, and NPK content in the material's eluate.

For determination of the maximum swelling degree ($S_{max}$) and swelling kinetics ($t_{63}$, time after 63% of $S_{max}$ is reached) were carried out by weighing an exact mass of about 0.1 g and placing it in two tea bags closed with cable ties. Hydrogel filled tea bags were placed in swelling medium (distilled water) and the swelling degree was determined gravimetrically after appropriate time intervals between 1.0 and 60.0 min. Swelling was determined using the following equations:

$$S_t = ((m_s - m_d)/m_d) \cdot 100 \ [\%];$$

$m_s$: swollen mass [g], $m_d$: dry mass of hydrogel [g].

For determination of the kinetic parameters mentioned above, the data were fitted into the Voigt model [23]:

$$S_t = S_{max} \times (1 - e^{-t/t63}).$$

Maximum swelling degrees were also determined in 0.9% NaCl and seawater solution to address salty conditions.

The AUL was determined by placing about 0.16 g of hydrogel in a cylinder with a porous bottom. A weight (0.3 psi) was placed on the gel and the cylinder was placed in a Petri dish filled with 50 mL water. The swelling degree was determined after 1 h. The AUL was different compared to free swelling without applied pressure on the gel. The application in soil was closer to AUL conditions.

Residual from the eluate was carried out gravimetrically by immersing 0.5 g of the sample in 500 mL water. After 24 h, the mixture was filtered and the hydrogel mass was determined after drying.

Nitrogen, phosphorous, and potassium in the eluate of these samples were analyzed after 24 h using Hach LCK cuvette tests (nitrogen: TNb, phosphorous: $PO_4^{3-}$-P and potassium: K).

The experiment was laid out using a completely randomized design with four replications and ten treatments. Each experimental unit was 2.1 m × 40 m = 84 m².

The tested factors were the following:

- Hydrogel types:

Arbovit® Polyacrylate (AP) (hydrogel mixed with clay minerals) and Huminsorb® Polyacrylate (HP) (hydrogel mixed with humic acids);

- Hydrogel application rates: 30 kg/ha and 20 kg/ha;
- Water supply rates: 100% and 50% of *ETc* crop evapotranspiration.

Two control experimental units were included without the application of hydrogels, with one receiving 100% *ETc* and the other receiving 50% *ETc* (Table 1). For statistical analysis, the first two factors were combined into one factor, representing the type and dose of hydrogel applied.

**Table 1.** The different applied treatments.

| Treatment | Type of Hydrogel | Hydrogel Application Rate | Water Intake Rate |
|---|---|---|---|
| T1 (AP, 20 kg, 50%) | Arbovit® Polyacrylate | 30 kg/ha | 100% *ETc* |
| T2 (AP, 30 kg, 50%) | | 20 kg/ha | 100% *ETc* |
| T3 (HP, 20 kg, 50%) | Huminsorb® Polyacrylate | 30 kg/ha | 100% *ETc* |
| T4 (HP, 30 kg, 50%) | | 20 kg/ha | 100% *ETc* |
| T5 (control, 50% *ETc*) | Without hydrogel | 0 kg/ha | 100% *ETc* |
| T6 (AP, 20 kg, 100%) | Arbovit® Polyacrylate | 30 kg/ha | 50% *ETc* |
| T7 (AP, 30 kg, 100%) | | 20 kg/ha | 50% *ETc* |
| T8 (HP, 20 kg, 100%) | Huminsorb® Polyacrylate | 30 kg/ha | 50% *ETc* |
| T9 (HP, 30 kg, 100%) | | 20 kg/ha | 50% *ETc* |
| T10 (control, 100% *ETc*) | Without hydrogel | 0 kg/ha | 50% *ETc* |

### 2.3. Irrigation Management

The irrigation method employed involved supplying the daily crop water requirement while considering the soil water content. Two water quantities representing a percentage of crop evapotranspiration (*ETc*) were compared: 100% *ETc* and 50% *ETc*. The water input was determined by accumulating the daily reference evapotranspiration values that define the climatic water demand ($ET_0$); they were corrected using the crop coefficient (*Kc*). The $ET_0$ value was obtained using a fully automated weather station (Adcon Device, Vienna, Austria) installed next to the test plot, and the recorded climatic data (temperature, relative humidity, solar radiation, and wind speed) were automatically integrated into the Penman–Monteith formula [24] in order to estimate the reference evapotranspiration. These data were sent automatically and continuously, with a frequency of 15 min, via GPRS (general packet radio service), facilitating the monitoring of the parameters and the calculation of the crop's water requirements.

The formula for calculating net irrigation requirements is

$$ETc = ET_0 \times Kc.$$

The crop coefficient of the onion crop used in this trial is the one published in the study by Lòpez-Urrea et al. [25]. The values of *Kc* according to the phenological stages are listed in the following table (Table 2).

**Table 2.** Onion crop coefficient values according to phenological stages.

| Stages | *Kc* |
|---|---|
| 1 leaf | 0.7 |
| 2 leaves | 0.8 |
| 3 to 4 leaves | 0.9 |
| Bulb formation | 1.05 |
| Bulb thickening | 1.20 |
| Bulb maturity | 0.7 |

*2.4. Observations and Data Recording*

- Non-destructive measurements

To monitor the growth of the onion plants, measurements were taken every 20 days on 6 samples for each repetition during the trial period. These measurements included the length and density of the leaves, collar diameter, bulb diameter, and leaf temperature. The relative growth rate (*RGR*) in $cm \cdot cm^{-1} \cdot d^{-1}$ was calculated using plant height measurements. Specifically, the height at time (t + n) was compared to the existing height at time t. The *RGR* was calculated using the following equation (adapted from James and Richards, [26]):

$$RGR = [Ln(\text{Height})_{t+n} - Ln(\text{Height})_t] / [(t + n) - t]$$

The *RGR* indicator was calculated based on plant height measurements taken every 20 days (n = 20).

Other measurements concerning leaf temperature were monitored with an infra-red thermometer (Trotec device, Heinsburg, Germany). The leaf temperature depends on the energy the leaf receives and eliminates.

Finally, the stomatal conductance was also studied on mature leaves using a porometer (Leaf PorometerDecagonDevices Inc., Pullman, WA, USA).

- Destructive measurements

Measurements of root mass and volume, fresh and dry biomass, and yield were taken for three samples in each replication immediately after harvest. Additionally, a biochemical analysis was conducted to determine proline levels as an evaluation parameter of stress level for each studied treatment. The extraction and determination of proline was performed according to the method described by Bates et al. [27]. For each treatment, samples of 100 mg (leaves and roots) were ground in 10 mL of 3% sulfosalicylic acid and then heated in a water bath at 85 degrees for one hour. The tubes are closed to avoid volatilization of the alcohol. After cooling, 1 mL of extract was taken and 1 mL of acetic acid ($CH_3COOH$), 25 mg ninhydrin and a 1 mL of mixture (12 mL distilled water, 30 mL acetic acid, and 8 mL orthophosphoric acid) was added. The tubes were incubated for 30 min at 100 degrees in a water bath. After turning progressively red, 4 mL of toluene were added to each tube after cooling. After shaking, two phases are formed, and the upper one containing proline was recovered. Finally, the optical density was determined using a spectrophotometer at a wavelength of 528 nm.

The concentration of proline analyzed in the leaves and roots of the plants is determined by referring to a standard line prepared from a standard range of proline based on the following concentrations: 5, 10, 15, 20, and 25 µg/mL of proline per tube.

- Statistical analysis

To compare the treatments, the standard analysis of variance (ANOVA) test was conducted using SPSS 25.0 statistical software [28]. If a significant difference was found, the means were compared using the Student–Newman–Keuls (SNK) test [29] at a 5% significance level. Furthermore, a contrast analysis was performed to investigate the differences between the various groups.

## 3. Results

### 3.1. Hydrogel Properties

Table 3 summarizes the results of the experiments carried out to investigate the differences between the hydrogels used.

**Table 3.** Properties of hydrogel-based products used for pot experiments.

| Sample | Arbovit | Huminsorb® |
|---|---|---|
| $S_{max}$ (distilled water) [%] | 13,883 | 4282 |
| $t_{63}$ [min] | 5.180 | 8.750 |
| $S_{max}$ (0.9% NaCl) [%] (% of swelling in distilled water) | 2339 (16.8) | 969 (22.6) |
| $t_{63}$ (0.9% NaCl) [min] | 11.6 | 25.7 |
| $S_{max}$ (seawater) [%] × (% of swelling in distilled water) | 641 (4.6) | 1238 (28.9) |
| $t_{63}$ (seawater) [min] | 37.4 | 38.9 |
| AUL (0.3 psi) [%] (% of free swelling) | 1986 (14.3) | 1179 (27.5) |
| Residual from eluate [%] | 4.6 | 30.8 |
| Nitrogen (TNb) [mg/L] | 2.79 | 0.95 |
| Phosphorous ($PO_4^{3-}$-P) [mg/L] | 0.41 | 0.30 |
| Potassium (K) [mg/L] | 59.33 | 62.30 |

The swelling degree is higher for the Arbovit® sample, although the hydrogel type in both samples is the same. Humic acid present in Huminsorb® is a water-soluble material (see residual from eluate). As in this set-up, the swelling is determined for the whole composite material gravimetrically; the lower swelling degree does not necessarily mean less water uptake in hydrogel. Rather, loss of mass due to dissolution of humic acid leads to lower weight of the sample upon swelling over the time. Swelling of hydrogels in salty media (0.9% NaCl and seawater), which was carried out to display extremely salty conditions, is much lower when compared to distilled water. Osmotic pressure is a main driving force for water uptake, which is limited upon increasing salt concentration. Upon further immersion in seawater, the hydrogels start to de-swell drastically over time. This is caused by the presence of bivalent cations. Swelling velocity—which can become an important factor upon water uptake during short irrigation times—is higher for Arbovit® and for AUL. Additionally, this can be explained by the solubility of the humic acid used as an additive in Huminsorb®. Characterization of the eluates' NPK-composition shows the most significant difference in the amount of total nitrogen (2.79 mg/L found in the eluate of Arbovit compared to 0.95 mg/L in the eluate of Huminsorb®).

### 3.2. Effect on Growth Parameters

The study revealed that both the type and dose of hydrogel, as well as the amount of water applied, significantly influenced the growth of onion plants as indicated by various vegetation parameters: number of leaves, plant height, and collar diameter. All three parameters displayed a consistent trend throughout the experiment. Figure 2 displays the growth curve of onion plants, measured as plant height, from day 21 after transplantation until the end of the study, for all treatments.

The analysis of plant height data showed a significant effect of both the hydrogel-type dose and water dose factors. Treatment T7, which combined the 100% dose and the AP hydrogel at 30 kg, resulted in the highest plant heights for all measurements, while the lowest plant heights were registered for the control treatment T5 (50% *ETc*). Although both types of hydrogels had a positive impact on plant height development, contrast analysis showed that the use of AP hydrogel yielded better results than HP hydrogel for both irrigation doses.

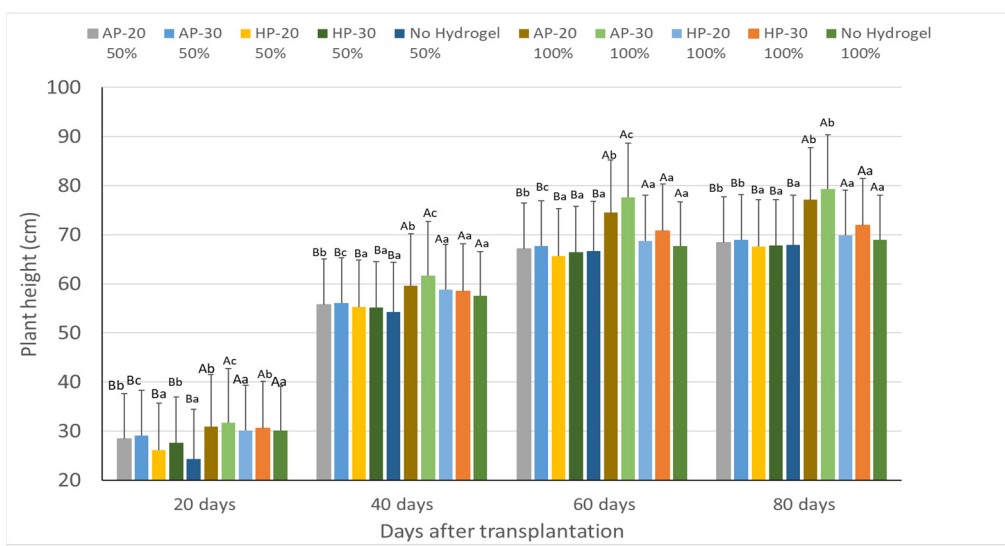

**Figure 2.** The evolution of onion plant height under the effect of different factors. Error bars represent standard deviations. Different letters A, B indicate significant differences ($p < 0.05$) between the treatments for the water dose factor and different letters a–c indicate significant differences ($p < 0.05$) between the treatments for the hydrogel type-dose factor.

This is consistent with the findings of previous studies by Sheikhmoradi et al. [30] and Yáñez-Chávez et al. [31] which demonstrated a significant increase in maize height with the highest hydrogel dose. Moreover, Abobatta [32] noted that hydrogel polymers improve plant growth by increasing the water-holding capacity of soil and delaying the wilting point, which can explain the results observed in this study. Overall, these findings highlight the importance of selecting the appropriate type and dose of hydrogel for enhancing plant growth and improving crop yield.

The effect of irrigation water amount was more evident on root development, with treatment T5 (50% $ETc$, no hydrogel) recording the highest root mass and volume at 2.57 g and 2.83 cm$^3$, respectively, as compared to treatment T7 (100% $ETc$ AP−30 kg/ha), which had the lowest values for both variables. This is in line with a study by Asseng et al. [33], which demonstrated that under water deficit conditions, total root development is significantly slowed down in the top 30 cm, while roots continue to grow in the deeper soil layers between 30 and 60 cm. After re-watering, the root growth pattern reverts to the fastest rates of root growth in the superficial soil layers. It is noteworthy that, overall, the root system grows more than the dry matter under water deficit conditions. This indicates that onion plants have the ability to adapt their root system to water availability in the soil, which can help them to survive and maintain growth under limited water conditions. Therefore, these findings suggest that careful irrigation management and the use of hydrogels can play a vital role in promoting onion plant growth and development.

The two variables mentioned (average root mass and average root volume) versus different treatments are shown in Figure 3a,b.

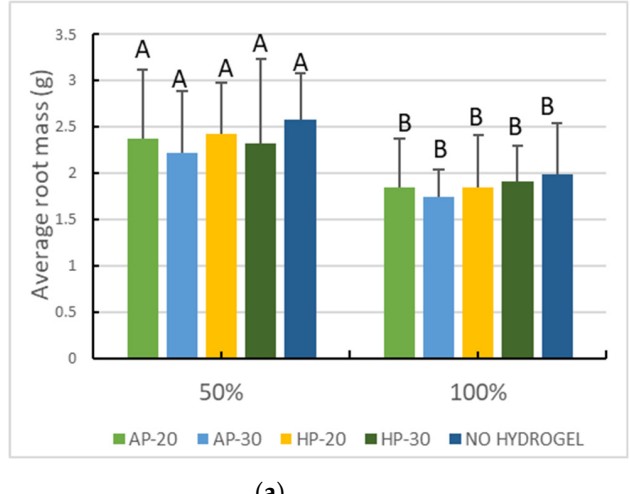
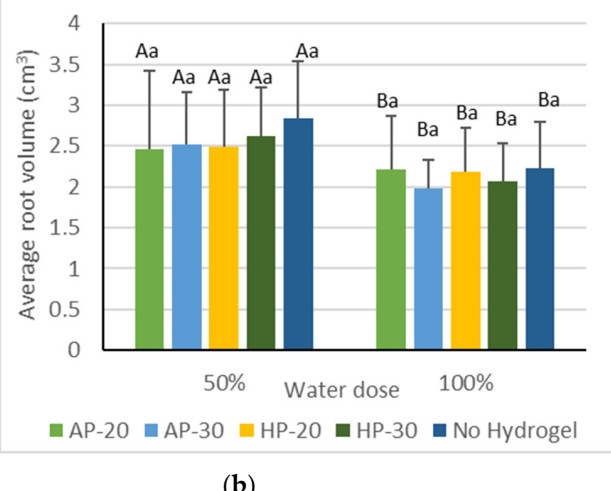

(**a**)  (**b**)

**Figure 3.** (**a**) Average root mass of the different treatments; (**b**) Average root volume of the different treatments. Error bars represent standard deviations. Different letters A, B indicate significant differences ($p < 0.05$) between the treatments for the water dose factor and different letters a–c indicate significant differences ($p < 0.05$) between the treatments for the hydrogel type-dose factor.

### 3.3. Effect on Eco-Physiological Parameters

All eco-physiological parameters (leaf temperature, stomatal conductance, and average chlorophyll index measurements) were significantly affected by both factors, except for leaf water content, which was only affected by the water dose and not the type or dose of hydrogel. In fact, the relative growth rate (*RGR*) was significantly affected by both factors, as shown in Figure 4, with treatments receiving hydrogel application having a higher growth rate than those without. The irrigation water dose had a greater impact, with treatments irrigated with 100% *ETc* showing better results than those irrigated with 50% *ETc*. Descriptive analysis of the data indicated that treatments T6 (100% *ETc*, AP, 20 kg/ha) and T7 (100% *ETc*, AP, 30 kg/ha) had the highest relative growth rates throughout the trial period, while treatment T5 (50% *ETc*, control) had the lowest values.

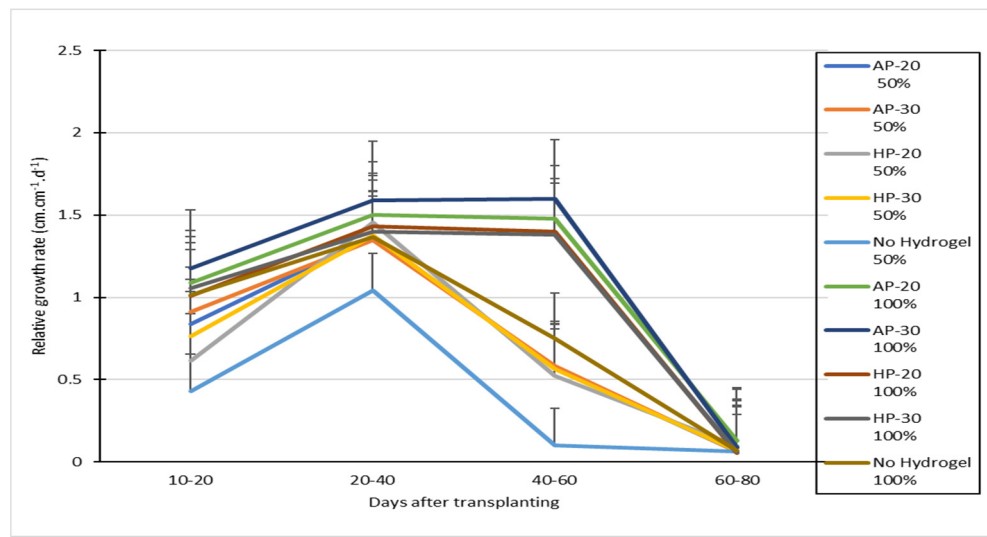

**Figure 4.** Evolution of onion relative growth rate under the effect of different factors. Error bars represent standard deviations.

However, from day 40 after transplantation, the trend of RGR evolution was completely reversed. Indeed, this rate continued to fall in all the plants except those of the 100% irrigated treatments which started to decrease only after day 60.

Notably, AP hydrogel produced better results than HP hydrogel, although both were beneficial to the plants. However, the difference between AP hydrogel and the control treatment was not significant.

The results of the analysis did not reveal a significant effect of the "hydrogel-type dose" factor on the plants' water content, while the amount of irrigation water had a significant effect. Based on the samples collected, it can be stated that treatments T9 (100% *ETc*, HP, 30 kg/ha) and T7 (100% *ETc*, AP, 30 kg/ha) had the highest water content values, at 89.87% and 89.86%, respectively, while treatments T3 (50% *ETc*, HP, 20 kg/ha) and T1 (50% *ETc*, AP, 20 kg/ha) had the lowest values, at 88.92% and 89.02%, respectively (Figure 5).

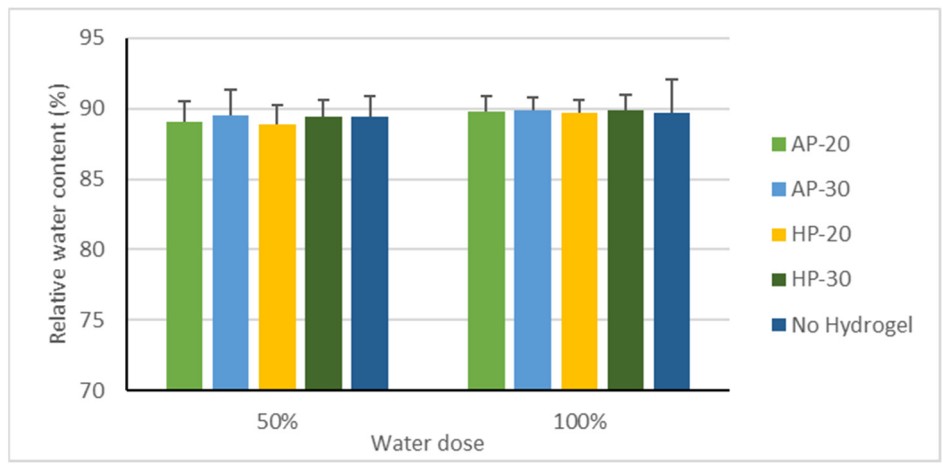

**Figure 5.** Water contents in the various treatments under the effect of the two factors. Error bars represent standard deviations.

Interestingly, the findings presented by Soubeih [34] indicate that the application of hydrogels led to a notable increase in the water content of onion plants in all tested samples when compared to the reference sample.

### 3.4. Effect on Bio-Chemical Parameters: Proline Content

Regarding the effect on proline, its highest concentration (0.097 $\mu mol \cdot g^{-1}$) was marked among the plants of treatment T5 (control, 50%), followed by the treatments that underwent a 50% *ETc* stress but with hydrogel use, i.e., T3 (dose = 50%, HP, 20 kg/ha), T4 (dose = 50%, HP, 30 kg/ha), T1 (dose = 50%, AP, 20 kg/ha), and T2 (dose = 50%, AP, 30 kg/ha). Nevertheless, it is important to highlight the fact that the combination of hydrogels with 50% *ETc* water content (treatment T1) allowed us to obtain values that were very close to those obtained by the control treatment T10 irrigated at 100% *ETc* (0.08 $\mu mol \cdot g^{-1}$) or even lower values than those for treatment T2 which achieved 0.078 $\mu mol \cdot g^{-1}$. This demonstrates the potential of the hydrogel technology to reduce the effect of stress on the plants and that even with a 50% reduction in water supply, the addition of the hydrogels created a favorable microclimate at the plants' roots and therefore reduced the stress. The unstressed treatments had minimal proline values (Figure 6). In fact, Hinojosa et al. [35] have highlighted the capacity of hydrogel to decrease soil permeability by increasing water retention and water absorption, which contribute to reducing the water stress of plants and may explain why hydrogels have been beneficial in treatments with water deficit.

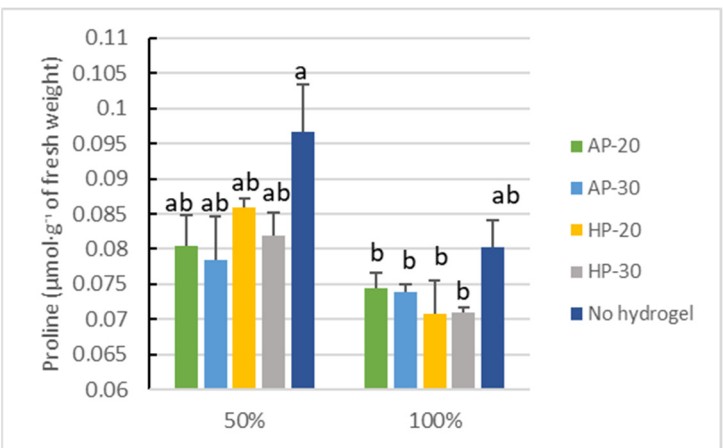

**Figure 6.** Effect of different irrigation treatments on proline content at the end of the cycle. Different letters a, b indicates significant differences ($p < 0.05$) between the treatments for the hydrogel type-dose factor.

Statistical analysis confirmed these results by revealing a significant effect of both factors on proline content, and the SNK test allowed the grouping of treatments as shown in Figure 6.

*3.5. Effect on Yield Parameters*

For the yield analysis, the descriptive statistics showed that there was no significant effect of the hydrogel-type dose factor or its interactions. However, the amount of irrigation water had a significant impact on the yield of the studied treatments. Treatments receiving 100% *ETc* irrigation had better yields when compared to those receiving only half the amount. It was also observed that the application of Huminsorb® as a hydrogel with a dose of 30 kg/ha resulted in the highest yields, reaching 70.5 t/ha (T9 (HP, 30 kg, 100%)). Conversely, the lowest yield was obtained among the treatments receiving 50% *ETc*, ranging from 56 t/ha to 47 t/ha, with the lowest yield recorded for treatment T5 (control, 50% d'*ETc*). Although the difference between the yields of treatments irrigated at 50% *ETc* was not significant, it should be noted that the addition of hydrogels resulted in an increase in yield of almost 9 t/ha. This could be explained by the fact that the use of hydrogels increases the potential yield of the crop by enhancing water availability within the root zone and reducing water and nutrient losses outside of it.

Despite this, the T4 treatment (50% *ETC*, HP, 30 kg/ha) produced a yield of 56 t/ha; this was almost equivalent to that of the control treatment T10 irrigated at 100% *ETc*, which yielded 59.7 t/ha (Figure 7). This indicates that even with a 50% reduction in crop water requirement, the addition of hydrogels did not negatively impact yield. This finding reinforces the idea that hydrogels enable significant water savings (in this case, 208 mm) while maintaining the same yield, resulting in increased crop water productivity.

In a similar vein, a field experiment conducted on loamy soil in India demonstrated that applying hydrogel at 5 kg/ha before lentil sowing resulted in a significant increase in seed yield by 51.1% (1032.1 kg/ha) when compared to the control [36]. Likewise, Bouaziz et al. [37] reported that potato yield significantly increased from 25.4% to 44.1% due to the use of different doses of hydrogel as compared to the control.

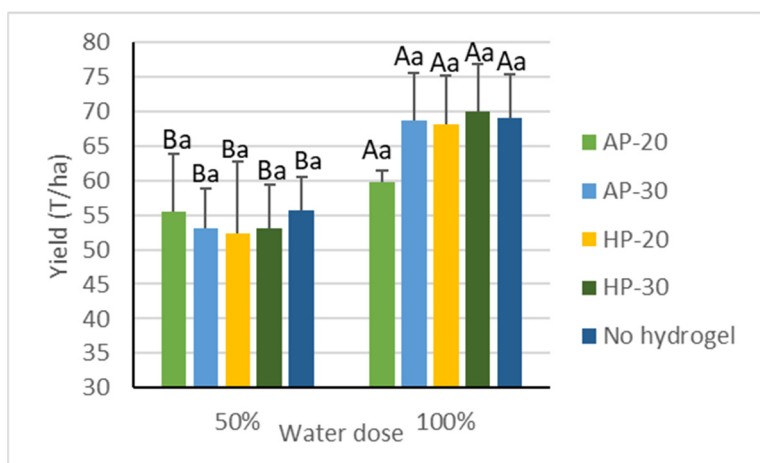

**Figure 7.** Yield per treatment in t/ha after drying out. Different letters A, B indicate significant differences ($p < 0.05$) between the treatments for the water dose factor and different letters a, b indicate significant differences ($p < 0.05$) between the treatments for the hydrogel type-dose factor.

The agronomic water productivity, known also by the crop water use efficiency, was calculated for the different studied treatments. The method of calculating agronomic water efficiency is based on its own definition. It is therefore the ratio of the produced fresh matter of the given crop to the volume of water received by the culture from rain and irrigation. Thus, agronomic water productivity ($kg/m^3$) = production/quantity of water was used [37].

Figure 8 shows the effect of the two factors on the agronomic water productivity.

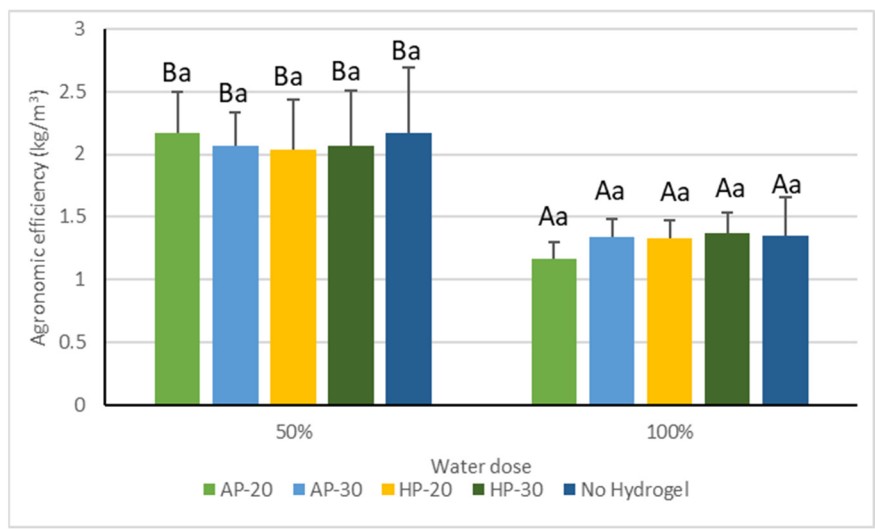

**Figure 8.** Effect of both factors on irrigation water use efficiency. Different letters A, B indicate significant differences ($p < 0.05$) between the treatments for the water dose factor and the same letter a indicate no significant difference at ($p < 0.05$) between the treatments for the hydrogel type-dose factor.

Figure 8 indicates that treatments receiving 50% *ETc* exhibited the highest crop water use efficiency (*WUE*) when compared to those receiving 100% *ETc*. The treatments with the highest *WUE* were T5 (control, 50%) and T1 (AP, 20 kg/ha, 50%), with values of 2.171 $kg/m^3$ and 2.165 $kg/m^3$, respectively.

However, it is important to note that the application of hydrogels and the subsequent reduction in water amount by 50% led to an increase in WUE, with a value of 2.165 $kg/m^3$

when compared to the control treatment with 100% *ETc* and a WUE of 1.3 kg/ha. This increase in WUE (by 0.8 kg/m$^3$) resulted in water savings of about 208 mm while improving water productivity without affecting the yield.

Our results are consistent with those of Albalasmeh et al. [38], who showed that *WUE* values increased with the concentration of the hydrogel and their effect on sandy soil was higher than on silty clay soil. Qin et al. [39] have shown that WUE was enhanced as a result of applying different types of hydrogels in potatoes as compared to no hydrogels.

Similarly, Asmar et al. [4], indicated that water use had increased once the hydrogel was used, as it reduced the amount of water infiltrated below the root zone.

These studies, as well as the present one, show how hydrogels can contribute to water resource management in Morocco to face the increasing scarcity of water resources along with poor piloting and absence of farmers' knowledge of their crops' needs. Thus, the country is in a phase of research of new alternatives. After having reconverted to localized irrigation and modernizing the collective irrigation networks, we can now think of adopting these modern technologies.

## 4. Conclusions

The application of hydrogels had a significant effect on all vegetative parameters, including number of leaves, plant height, stem diameter, root mass, and volume. Similar results were observed for ecophysiological parameters, such as leaf temperature and RGR. Arbovit® Polyacrylate hydrogel application yielded better results in aerial vegetative growth, particularly at a dosage of 30 kg per hectare and under full irrigation (100% *ETc*). It is noteworthy that using AP hydrogel at both 20 kg/ha and 30 kg/ha under 50% *ETc* resulted in similar agronomic performance in terms of yield and production parameters. Furthermore, AP hydrogel application at both dosages led to better plant growth, as observed in the relative growth rate.

The study also revealed that water dosage had a significant effect on average bulb weight and final marketable bulb yield. Huminsorb® Polyacrylate hydrogel amendment resulted in better yields: 70.55 t/ha for the T9 treatment (100%, HP, 30 kg/ha) with a gain of 11 t/ha when compared to the control treatment with 100% *ETc* without hydrogel application. Combining hydrogel with a 50% *ETc* regime (T4 (HP, 30 kg, 50%)) led to water savings of 208 mm while still achieving almost the same yield as obtained with 100% *ETc* without hydrogel. The lowest yields were recorded for treatments with 50% *ETc* application (control, 50%) and no hydrogel application, reaching only 47.58 t/ha.

Finally, the highest agronomic efficiencies were observed for treatments receiving 50% crop water evapotranspiration, reaching 2.1 kg/m$^3$ as compared to those with full irrigation and a WUE of 1.3 kg/m$^3$. In conclusion, the study demonstrated that hydrogel application led to improved agronomic performance, particularly under water stress conditions, while also providing water savings and enhancing water productivity. This was also proven by the proline content results where the stressed treatments with hydrogel amendments gave similar results to the 100% *ETc* irrigated control treatment, unlike the 50% *ETc* irrigated control treatment which had a higher proline value.

Based on the results of this study, the use of Arbovit® Polyacrylate hydrogel with a dose of 30 kg/ha under full irrigation conditions (100% *ETc*) is recommended to achieve the best aerial vegetative growth. For water-limited conditions (50% *ETc*), the use of hydrogels either with a dose of 20 or 30 kg/ha can lead to similar agronomic performance in terms of yield and production parameters when compared to full irrigation without hydrogel. However, the use of hydrogels should be combined with appropriate irrigation management practices to achieve optimal results in terms of water productivity and yield. Further research is needed to evaluate the long-term effects of hydrogel applications on soil properties and crop performance. More research and development of new hydrogels with good biodegradability and based on renewable resources is needed as well. Development of hydrogels adapted to specific soil conditions could combine water saving with a supply of specially needed nutrients. Additionally, comparative studies with products newly

available in the market need to be carried out. These research activities are crucial to help to deal with drought and water scarcity, which are increasing problems in Morocco. The significance of such research will continue to grow in importance for regions currently experiencing the effects of climate change.

**Author Contributions:** Conceptualization, A.A. (Aziz Abouabdillah); methodology, A.A. (Aziz Abouabdillah), O.E.B. and A.A. (Abdellah Aboudrare); writing—original draft, O.E.B., A.A. (Aziz Abouabdillah) and M.B. (Mohamed Bourioug); writing—review and editing, R.B., M.K., D.S., M.B. (Markus Biel) and S.R.; validation, R.B., M.F. and J.M.; funding acquisition: M.K. and A.J. All authors have read and agreed to the published version of the manuscript.

**Funding:** As a part German Moroccan project I-WALAMAR financed by the Ministry of Higher Education and Research in Germany, this work was supervised by the Research Institute of Water Management and Climate future, RWTH Aachen University. The opinions expressed in this publication are those of the authors and do not necessarily reflect the views of the FIW RWTH.

**Data Availability Statement:** Not applicable.

**Acknowledgments:** The authors would like to express their sincere gratitude to the Ministry of Higher Education and Research in Germany for financing the I-WALAMAR project, a collaborative project between Moroccan and German partners. We extend our heartfelt thanks to all the German and Moroccan partners for technical support throughout the project. We are grateful to the University Moulay Ismail of Meknes for their collaboration and support during the project. Special thanks to Imane Taibi, a master's student, for her hard work in crop installation, monitoring, and data analysis. Lastly, we acknowledge the National School of Agriculture of Meknes for providing us with the necessary support and logistics to conduct this study.

**Conflicts of Interest:** The authors declare no conflict of interest.

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
