# Peer review of "Innovative Solutions for Drought: Evaluating Hydrogel Application on Onion Cultivation (Allium cepa) in Morocco"

_water, doi:10.3390/w15111972_

Round 1
Reviewer 1 Report
Hydrogel technology is a significant approach to enhance water use efficiency in agricultural plants, thereby conserving water resources and augmenting plant drought resistance. This study assesses the impact of hydrogel application on onion (Allium cepa) cultivation in Meknes, Morocco, by comparing two distinct types of hydrogels, Arbovit polyacrylate, and Humisorb polyacrylate. The research findings hold good theoretical reference value. Nevertheless, this article needs improvement in the following areas:
(1) The manuscript requires formatting and correction of language, wording, and grammatical errors to align with the journal's requirements.
(2) The introduction should add a historical rainfall and drought analysis of the study region to help readers understand the significance of the research.
(3) The article requires supplementary calculations and discussions on the hydrogel technology's water-saving effect and its potential contribution to water resource management in Morocco.
(4) The conclusion should discuss the potential challenges, problems, or limiting factors of hydrogel technology in practical applications. Moreover, it should highlight the technical advantages of this study compared to existing hydrogel technology.
(5) The English language and style are acceptable, with minor spell checks required.
In conclusion, the manuscript requires revision and review to address the above concerns.
Hydrogel technology is a significant approach to enhance water use efficiency in agricultural plants, thereby conserving water resources and augmenting plant drought resistance. This study assesses the impact of hydrogel application on onion (Allium cepa) cultivation in Meknes, Morocco, by comparing two distinct types of hydrogels, Arbovit polyacrylate, and Humisorb polyacrylate. The research findings hold good theoretical reference value. Nevertheless, this article needs improvement in the following areas:
(1) The manuscript requires formatting and correction of language, wording, and grammatical errors to align with the journal's requirements.
(2) The introduction should add a historical rainfall and drought analysis of the study region to help readers understand the significance of the research.
(3) The article requires supplementary calculations and discussions on the hydrogel technology's water-saving effect and its potential contribution to water resource management in Morocco.
(4) The conclusion should discuss the potential challenges, problems, or limiting factors of hydrogel technology in practical applications. Moreover, it should highlight the technical advantages of this study compared to existing hydrogel technology.
(5) The English language and style are acceptable, with minor spell checks required.
In conclusion, the manuscript requires revision and review to address the above concerns.
Reviewer 2 Report
General comments
The manuscript evaluates the results of a study on the impact of hydrogel application on onion (Allium cepa) cultivation in Meknes, Morocco. The authors have invested a great deal of time and effort in organising their study, which examines the use of hydrogels to combat water scarcity in an arid region of Morocco. The subject of the study is interesting and probably to be relevant to other countries in arid and semi-arid regions. However, the manuscript is not well written. It needs enough work to make it publishable. See the comments below for details
Specifically,
1. The manuscript must be written according to the Instructions for Authors of the Journal. See website for details https://www.mdpi.com/journal/water/instructions and also, the “Reference List and Citations Style Guide for MDPI Journals” at the website: https://www.mdpi.com/authors/references.
2. References are not in the correct format for this journal. Pay attention to the Journal name, which should be in italic and abbreviated form (see instructions for authors).
3. A general check of the references is needed (References list).
4. English needs quite a bit of work, both in grammar and in wording. Perhaps working with an independent English-speaking scholar would help in the improving of the text.
5. Italics should be used for variables in the main text, tables and equations.
6. Figures in higher analysis.
7. The Introduction of the manuscript is weak and needs improvement. In particular, it should (a) place the study in a broader context and emphasize why it is important, (b) describe its objectives, (c) define the purpose of the study and its significance, (d) carefully examine the current state of the research field and the main publications cited, (e) summarize the main objective of the study, and finally, (f) highlight the main conclusions. In the present case, the aim of the study is omitted.
8. A map of the wider area showing the location of the experimental site must be included in subsection entitled "2.1 Experimental Site"
9. Clarify how crop evapotranspiration (ETc) is calculated in your study.
10. Throughout the main text, the authors use "T" as the symbol for the metric ton, whereas in the International System of Units (SI) uses "t", not "T".
Specific comments
Line 57
“Additionally, [10] and [11] observed a stimulating…”
should be
“Additionally, Tyagi et al. [10] and Hou et al. [11] observed a stimulating…”
Line 61
“According to [12], the use of hydrogels…”
should be
“According to Johnson [12], the use of hydrogels…”
Line 70
“Following [13], polymers can…”
should be
“Following Decriaud-Calmon et al. [13], polymers can…”
Lines 122, 125, Table 2
The authors write elsewhere “t_63” (Line 122), elsewhere “t63” (Line 125) and elsewhere “t63” (Table 2).
COMMENT: Keep the same symbol throughout the main text.
Line 151
“The table below displays the various treatments applied:”
should be
“Table 1 shows the different treatments applied”
Line 154
“The irrigation method employed in this study involved supplying……”
should be
“The irrigation method employed involved supplying……”
Line 162
Define the abbreviation “GPRS”
Line 169
This following expression is not understood” “cm.cm-1.D-1”
Line 170
“…the height at time (t + n) was compared…”
COMMENT: Define the symbol “n”
Lines 171 – 172
Add the citation “James and Richards, 2006” in the Reference List.
Line 177 – 179
“Additionally, a biochemical analysis was conducted to determine proline levels as an evaluation parameter for stress level for each studied treatment.”
COMMENT: More information on biochemical analysis is required.
Line 182
“…. using SPSS 25.0 (COMMENT: Add citation) statistical….”
Line 183
“….using the Student Newman and Keuls (SNK) test (COMMENT: Add citation)…”
Line 189
“The table below summarizes…”
should be
“Table 2 summarizes…”
Line 192
Please include the following note in the main text: “*upon further immersion in seawater, the hydrogels de-swell drastically”
Line 223
“….studies by [15] and [16], which….”
should be
“….studies by Sheikhmoradi et al. [15] and Yáñez-Chávez et al. [16], which….”
Lines 224 – 225
“Moreover, [17] noted that….”
should be
“Moreover, Abobatta [17] noted that….”
Line 233
“…a study by [18], which….”
should be
“…a study by Asseng et al. [18], which….”
Line 244
“The figures below present the two mentioned variables.”
should be
“The two variables mentioned (average root mass and average root volume) versus different treatments are shown in Figures 2a, b”.
Figure 2a
Y-axis: “Root mass”; Figure caption: “Average root mass”
Figure 2b
Y-axis: “Root volume”; Figure caption: “Average Root volume”
COMMENT: Check and correction.
Line 247
The authors refer to eco-physiological parameters but do not specify them. What exactly does this term mean and what does it cover?
Figure 3, y-axis
The following is not understood: “(cm.cm-1.j-1)”.
Line 299
“Figure 5: Yield per treatment in tons per hectare after drying out”
should be
“Figure 5: Yield per treatment in t/ha after drying out”
Line 302
“Likewise, [11] reported….”
should be
“Likewise, Hou et al. [11] reported….”
Lines 305 - 306
“The agronomic water productivity, known also by the crop water use efficiency was calculated for the different studied treatments”.
COMMENT: Please clarify and provide more information on the calculation of 'agronomic water productivity'. Also, agronomic water productivity appears firstly in lines 305 and 306, and its calculation follows in lines 308 and 309.
Line 312
“The above graph indicates….”
should be
“Figure 6 indicates….”
Lines 415 – 416
“Abobatta, W. (Éd.). Impact of hydrogel polymer in agricultural sector. Advances in Agriculture and Environmental Science: 415 Open Access (AAEOA), (2018). 1(2), 59-64”
should be
“Abobatta, W. Impact of hydrogel polymer in agricultural sector. Adv Agr Environ Sci 2018, 1(2), 59−64. DOI: 10.30881/aaeoa.00011”
The manuscript evaluates the results of a study on the impact of hydrogel application on onion (Allium cepa) cultivation in Meknes, Morocco. The authors have invested a great deal of time and effort in organising their study, which examines the use of hydrogels to combat water scarcity in a arid region of Morocco. The subject of the study is interesting and probably to be relevant to other countries in arid and semi-arid regions. However, the manuscript is not well written. It needs enough work to make it publishable. Therefore, I recommend acceptance of the manuscript after major revisions. For details please, see my Recommendations for Authors.
Round 2
Reviewer 1 Report
I read the manuscript carefully, and I think it can be published
I read the manuscript carefully, and I think it can be published
Reviewer 2 Report
General comments
This is the second time I have reviewed this manuscript. Last time I had found serious problems (see my review report) and had suggested acceptance after major revisions. The authors have accepted all the comments and have made a serious and worthwhile further effort to improve the manuscript considerably. However, further efforts are still necessary for this manuscript to be published. See the comments below.
Specifically,
1. The manuscript must be written according to the Instructions for Authors of the Journal. See website for details https://www.mdpi.com/journal/water/instructions and also, the “Reference List and Citations Style Guide for MDPI Journals” at the website: https://www.mdpi.com/authors/references.
2. References are not in the correct format for this journal (see instructions for authors).
3. A general check of the references is needed (References list).
4. English needs quite a bit of work, both in grammar and in wording.
5. Italics should be used for variables in the main text, tables and equations.
6. Figures are not clear. They should be of a higher quality (higher analysis).
7. Figure 1 is protected by copyright?
Specific comments
Line 22
“… daily crop evapotranspiration, "ETc"). Two control…”
should be
“… daily crop evapotranspiration, ETc). Two control…”
Line 47
Define the abbreviation “GDP”
Lines 82, 529
Authors state “Satriani et al. [17]” (Line 87) and “[17] Atriani, A.” (Line529)
COMMENT: Please check. "Satriani" or "Atriani"?
Line 121
“…Meknes (figure 1). The…”
should be
“…Meknes (Figure 1). The…”
Line 124
“Figure 1 : location of the experiment site on Morocco’s map (from [23])”
should be
“Figure 1. Location of the experiment site on Morocco’s map [23]”
Line 125
“…an altitude of 516 meters…”
should be
“…an altitude of 516 m…”
Lines 132 – 133
“The experimental material investigated in this study comprised of locally sourced red variety onion crop …”
should be
“The experimental material investigated comprised of locally sourced red variety onion crop …”
Lines 134, 169, 172, 233, 235, 236, 238, 243
“ml” should be “mL”
COMMENT: In the International System of Units (SI), the symbol for the liter is “l”. However, in order to avoid confusion between the letter l (el) and the number 1 (one), a capital L has been adopted as an alternative symbol.
Line 161
“md: dry mass of hydrogel (COMMENT: Add units)
Lines 190, 191
The authors write “Table 1 shows the different treatments applied:” (Line 190) and immediately afterwards “Table 1: Different studied treatments” (Line 191)
COMMENT: Rewording correctly both.
Line 196
Define the abbreviation “ETâ‚€”
Lines 196, 205, 208
The authors refer to the symbol Kc elsewhere as “crop coefficient” and elsewhere as “cultural coefficient” (Lines 205, 208).
COMMENT: Keep the same term throughout the main text.
Line 199
“…the Penman-Monteith formula (COMMENT: Add citation) using
Lines 199 – 200
“…. the Penman-Monteith formula using a calculation software integrated in the weather station”.
COMMENT: More information about this software should be provided.
Line 214
“cm.cm-1.D-1”
should be
“cm.cm-1.d-1”
Line 232
“Bates et al.”
should be
“Bates et al. [27]”
Lines 231 – 234
“Extraction and determination of proline was performed according to the method described by Bates et al. v 100 mg of leaf and root samples from each treatment were ground in 10 ml of 3% sulfosalicylic acid and then heated in an 85 degree water bath for one hour”.
COMMENT: This text is unclear. It needs to be carefully rewritten in correct English.
Line 318
“…..measurements) in this study were significantly….”
should be
“…..measurements) were significantly….”
Line 342
“….by Soubeih Kh. [34] indicate…”
should be
“….by Soubeih [34] indicate
Line 498
“Water 2021, Volume 13, doi:10.3390/w13162167”
should be
“Water 2021, 13(16), 2167; doi:10.3390/w13162167”
Line 500
“Regional Environmental Change”
should be
“Reg. Environ. Change”
Line 501
“Journal of Climate”
should be
“J. Clim.”
Lines 516 – 517
“Journal of Soil Science and Environmental Management”
should be
“J. Soil Sci. Environ. Manage.”
Line 522
“Journal of Applied Biosciences”
should be
“J. Appl. Biosci.”
Line 524
“Journal of Pharmacognosy and Phytochemistry”
should be
“J. Pharmacognosy Phytother.”
Line 527
“Journal of the Science of Food and Agriculture”
should be
“J. Sci. Food Agric.”
Lines 530, 548
“Agricultural Water Management”
should be
“Agric. Water Manag.”
Line 551
“Journal of Ecology”
should be
“J. Ecol.”
Lines 561 - 562
“Journal of Agriculture and Environmental Sciences”
should be
“J. Agric. Environ. Sci.”
Line 569
“Soil Biology and Biochemistry”
should be
“Soil Biol. Biochem.”
Line 576
“Journal of the Saudi Society of Agricultural Sciences”
should be
“J. Saudi Soc. Agric. Sci.”
General comments
This is the second time I have reviewed this manuscript. Last time I had found serious problems (see my review report) and had suggested acceptance after major revisions. The authors have accepted all the comments and have made a serious and worthwhile further effort to improve the manuscript considerably. However, further efforts are still necessary for this manuscript to be published. See the comments below.
Specifically,
1. The manuscript must be written according to the Instructions for Authors of the Journal. See website for details https://www.mdpi.com/journal/water/instructions and also, the “Reference List and Citations Style Guide for MDPI Journals” at the website: https://www.mdpi.com/authors/references.
2. References are not in the correct format for this journal (see instructions for authors).
3. A general check of the references is needed (References list).
4. English needs quite a bit of work, both in grammar and in wording.
5. Italics should be used for variables in the main text, tables and equations.
6. Figures are not clear. They should be of a higher quality (higher analysis).
7. Figure 1 is protected by copyright?
Specific comments
Line 22
“… daily crop evapotranspiration, "ETc"). Two control…”
should be
“… daily crop evapotranspiration, ETc). Two control…”
Line 47
Define the abbreviation “GDP”
Lines 82, 529
Authors state “Satriani et al. [17]” (Line 87) and “[17] Atriani, A.” (Line529)
COMMENT: Please check. "Satriani" or "Atriani"?
Line 121
“…Meknes (figure 1). The…”
should be
“…Meknes (Figure 1). The…”
Line 124
“Figure 1 : location of the experiment site on Morocco’s map (from [23])”
should be
“Figure 1. Location of the experiment site on Morocco’s map [23]”
Line 125
“…an altitude of 516 meters…”
should be
“…an altitude of 516 m…”
Lines 132 – 133
“The experimental material investigated in this study comprised of locally sourced red variety onion crop …”
should be
“The experimental material investigated comprised of locally sourced red variety onion crop …”
Lines 134, 169, 172, 233, 235, 236, 238, 243
“ml” should be “mL”
COMMENT: In the International System of Units (SI), the symbol for the liter is “l”. However, in order to avoid confusion between the letter l (el) and the number 1 (one), a capital L has been adopted as an alternative symbol.
Line 161
“md: dry mass of hydrogel (COMMENT: Add units)
Lines 190, 191
The authors write “Table 1 shows the different treatments applied:” (Line 190) and immediately afterwards “Table 1: Different studied treatments” (Line 191)
COMMENT: Rewording correctly both.
Line 196
Define the abbreviation “ETâ‚€”
Lines 196, 205, 208
The authors refer to the symbol Kc elsewhere as “crop coefficient” and elsewhere as “cultural coefficient” (Lines 205, 208).
COMMENT: Keep the same term throughout the main text.
Line 199
“…the Penman-Monteith formula (COMMENT: Add citation) using
Lines 199 – 200
“…. the Penman-Monteith formula using a calculation software integrated in the weather station”.
COMMENT: More information about this software should be provided.
Line 214
“cm.cm-1.D-1”
should be
“cm.cm-1.d-1”
Line 232
“Bates et al.”
should be
“Bates et al. [27]”
Lines 231 – 234
“Extraction and determination of proline was performed according to the method described by Bates et al. v 100 mg of leaf and root samples from each treatment were ground in 10 ml of 3% sulfosalicylic acid and then heated in an 85 degree water bath for one hour”.
COMMENT: This text is unclear. It needs to be carefully rewritten in correct English.
Line 318
“…..measurements) in this study were significantly….”
should be
“…..measurements) were significantly….”
Line 342
“….by Soubeih Kh. [34] indicate…”
should be
“….by Soubeih [34] indicate
Line 498
“Water 2021, Volume 13, doi:10.3390/w13162167”
should be
“Water 2021, 13(16), 2167; doi:10.3390/w13162167”
Line 500
“Regional Environmental Change”
should be
“Reg. Environ. Change”
Line 501
“Journal of Climate”
should be
“J. Clim.”
Lines 516 – 517
“Journal of Soil Science and Environmental Management”
should be
“J. Soil Sci. Environ. Manage.”
Line 522
“Journal of Applied Biosciences”
should be
“J. Appl. Biosci.”
Line 524
“Journal of Pharmacognosy and Phytochemistry”
should be
“J. Pharmacognosy Phytother.”
Line 527
“Journal of the Science of Food and Agriculture”
should be
“J. Sci. Food Agric.”
Lines 530, 548
“Agricultural Water Management”
should be
“Agric. Water Manag.”
Line 551
“Journal of Ecology”
should be
“J. Ecol.”
Lines 561 - 562
“Journal of Agriculture and Environmental Sciences”
should be
“J. Agric. Environ. Sci.”
Line 569
“Soil Biology and Biochemistry”
should be
“Soil Biol. Biochem.”
Line 576
“Journal of the Saudi Society of Agricultural Sciences”
should be
“J. Saudi Soc. Agric. Sci.”
